# Advance and Challenges in the Treatment of Skin Diseases with the Transdermal Drug Delivery System

**DOI:** 10.3390/pharmaceutics15082165

**Published:** 2023-08-21

**Authors:** Tingting Cheng, Zongguang Tai, Min Shen, Ying Li, Junxia Yu, Jiandong Wang, Quangang Zhu, Zhongjian Chen

**Affiliations:** 1School of Pharmacy, Bengbu Medical College, 2600 Donghai Road, Bengbu 233030, China; ctt7881@163.com (T.C.); yjx17856271969@163.com (J.Y.); wjd0028@163.com (J.W.); 2Shanghai Skin Disease Hospital, School of Medicine, Tongji University, 1278 Baode Road, Shanghai 200443, China; taizongguang@126.com (Z.T.); shenmin0924@126.com (M.S.); rsrs1026@163.com (Y.L.)

**Keywords:** advance, bioequivalence, challenge, skin disease, transdermal drug delivery system

## Abstract

Skin diseases are among the most prevalent non-fatal conditions worldwide. The transdermal drug delivery system (TDDS) has emerged as a promising approach for treating skin diseases, owing to its numerous advantages such as high bioavailability, low systemic toxicity, and improved patient compliance. However, the effectiveness of the TDDS is hindered by several factors, including the barrier properties of the stratum corneum, the nature of the drug and carrier, and delivery conditions. In this paper, we provide an overview of the development of the TDDS from first-generation to fourth-generation systems, highlighting the characteristics of each carrier in terms of mechanism composition, penetration method, mechanism of action, and recent preclinical studies. We further investigated the significant challenges encountered in the development of the TDDS and the crucial significance of clinical trials.

## 1. Introduction

The epidermis, dermis, and subcutaneous tissues make up the skin, the biggest organ in the body, which also comprises auxiliary organs such as hair follicles and sebaceous glands (Figure 1). In a healthy state, the skin acts as a natural barrier to maintain a stable environment both within and outside the body. However, when the skin is diseased, the barrier function is compromised, small molecules and microorganisms can easily pass through the stratum corneum, and vital nutrients, including water and electrolytes, are easily lost from the body, upsetting this balance and leading to disease. The environment, stress, sleep disorders, and many other parameters are just a few factors that might harm the skin barrier. As society continues to change and people’s lifestyles and the environment change, skin disorders have become more common in recent years. With a prevalence rate of 25%, skin conditions are currently the fourth most prevalent non-fatal disease around the globe [1,2]. In severe situations, patients feel considerably sick, and even their jobs and personal lives are negatively impacted.

The two main treatment options for skin diseases are medication and physical therapy. Transdermal administration, as opposed to oral and injectable medications, has the advantage of being delivered directly to the damaged skin, as skin disorders manifest primarily as skin lesions exposed to the body surface, although all the above techniques can exert therapeutic effects. The transdermal drug delivery system (TDDS) is widely used and highly rated by researchers. According to previous studies, more than 70% of patients and physicians select the TDDS for dermatological diseases [3]. This is mainly because the TDDS has unique advantages over oral and topical therapies, including (1) direct effects on the skin, avoiding the first-pass effect in the liver, and high bioavailability; (2) maintenance of a constant blood concentration without peak and valley effects; (3) low incidence of systemic adverse effects; (4) high patient compliance; and (5) the ability to swiftly discontinue drug administration. TDDSs still hardly achieve the expected efficacy, and their disadvantages mainly comprise (1) a limited permeation efficiency, (2) limited tissue concentration maintenance duration, (3) high dose frequency, and (4) rapid skin irritation. Thanks to rapidly advancing drug delivery technology, the efficiency of medication penetration and precision drug distribution has considerably increased in the past few decades.

This review focuses on the chronological development of the TDDS, examines the history of TDDS development methodically, and evaluates the variables affecting TDDS penetration and the hardships encountered at this point.

## 2. Absorption Process of TDDS

### 2.1. The Process of Absorption

The trans-epidermal (stratum corneum) pathway and the trans-appendage pathway are the two types of TDDS penetration in the skin. The stratum corneum, the outermost layer of the skin, is the factor with the greatest impact on the skin penetration of medicines [4]. Its structure has been compared to “brick and mortar” because it is not uniform. The “bricks” are mainly keratinocytes, while the “mortar” refers to the highly hydrophobic lipids surrounding the keratinocytes, which are tightly bound together to form a natural barrier.

The stratum corneum pathway is further classified as intracellular or intercellular. The term “intercellular lipid route” refers to drug delivery via keratinocytes, which is the route utilized by most medicines. However, this route requires pharmaceuticals to traverse lipophilic and hydrophilic barriers, which may quickly result in the development of drug resistance. The transcellular pathway involves transport via the interstitial space of keratinocytes, but since the stratum corneum is so densely organized and irregular, drug molecules must transit via a constrained and convoluted channel, which impedes drug transport [5]. Trans-appendage route transport not only aids in the targeted transport of medications to deeper portions of the skin but also acts as a reservoir to increase the amount of drug stored, thereby extending the concentration gradient and boosting passive drug transport efficiency [6]. However, because cutaneous appendages only make up a small portion of the human skin surface area (about 0.1%), the appendage pathway has historically been given less weight [7]. The appendage pathway, however, has recently emerged as a research hotspot in the transport of nanocarriers due to mounting evidence of its significant role in skin penetration, particularly for nanocarriers [8].

### 2.2. Factors Affecting TDDS Penetration

The factors influencing medication transdermal penetration can be classified into three categories: physiological factors, drug physicochemical qualities, and delivery settings.

#### 2.2.1. Physiological Factors

The temperature and health of the skin are important factors in TDDS transdermal permeation [9]. When the temperature rises, subcutaneous blood vessels dilate and the permeability coefficient rises; moist skin hydrates and loosens the stratum corneum, enhancing medication penetration. Furthermore, race, gender, age, and location alter the thickness, water content, and blood flow of the stratum corneum, all of which influence medication transdermal delivery.

#### 2.2.2. Drug Physicochemical Qualities

According to Fick’s first law: J = k × ∆c = (D × ∆c × Kp)/I, the release of a drug is connected to the characteristics of the drug’s diffusion coefficient, partition coefficient, and concentration gradient. The higher the concentration gradient and the larger the diffusion and partition coefficients, the more advantageous the drug molecules are for transport. Small molecules of lipophilic medications are generally permeable, but big molecules of hydrophilic pharmaceuticals are not [10].

#### 2.2.3. Delivery Settings

The dosage form, matrix, pH, delivery concentration, and area of administration of the formulation are all important aspects in determining the drug’s transdermal penetration. In different matrices, the rate of drug absorption is emulsified > animal fats > lanolin > vegetable oils > hydrocarbons, and drug penetration is preferred when the pH of the matrix is less than the pKa of acidic medications or more than the pKa of alkaline drugs. The greater the drug concentration, the larger the region of administration, and the greater the penetration effectiveness [11].

## 3. Development of the TDDS

The stratum corneum, pharmacological characteristics, carrier, temperature, and pH all impact the transdermal penetration of the TDDS, reducing its efficacy. Investigators have supported the development of the TDDS and have continued to refine the transport carriers for the TDDS in an effort to increase permeation efficiency. TDDSs have gone through four stages of evolution since their introduction in 1981 [12]. Though drugs that may be applied as patches are rare, they are routinely combined with other permeation-promoting techniques in the first-generation TDDS. Most second-generation TDDS carriers use the techniques of sonophoresis, iontophoresis, prodrugs, chemical permeability enhancers, etc., which increase drug penetration but may cause irreversible physiological damage; in the third-generation TDDS, investigators have increased their research on electroporation and microneedles; the latest-generation TDDS largely refer to nanocarriers, which are also the most extensively evaluated category of carriers in the field of TDDS currently [13]. These unique carriers make it possible for hydrophilic biomolecules with low transdermal penetration to be transported across membranes, helping drugs penetrate deeper skin layers.

### 3.1. Frist-Generation TDDS

The first generation of TDDSs consisted mainly of skeletonized transdermal patches. The scopolamine patch was the pioneering transdermal patch approved by the Food and Drug Administration (FDA) in 1979 for the treatment of motion sickness [14]. The first generation of TDDS is based primarily on natural drug diffusion, which is only suitable for a few drugs due to the barrier effect of the stratum corneum and only drug molecules that meet the requirements formulated by Lipinski’s Rule 5, those with a log *p*-value of 1–3 and molecular weights of 500 Da [15]. Transdermal patches, including scopolamine, nicotine, and nitroglycerin patches, are frequently used in therapeutic settings and may also be utilized with alternative delivery methods [16,17].

### 3.2. Second-Generation TDDS

The second generation of TDDS aids drug passage through the skin by breaking the stratum corneum’s barrier function or by providing some form of pushing force for the drug molecules. This “disruption” is reversible and does not harm skin tissue. Second-generation TDDS is confined to small lipophilic compounds and has no influence on large or hydrophilic molecule skin penetration. Iontophoresis, premedication, chemical penetration enhancers, and ultrasound are the major components of second-generation TDDS. The second generation of TDDS is described in Table 1.

#### 3.2.1. Iontophoresis

Iontophoresis is a second-generation TDDS that uses a mild low-voltage current (10 volts or less) to accurately control drug transport through the stratum corneum, even into the systemic circulation [21]. Electro-rejection (ER) and electro-osmosis (EO), a treatment dating back to the 18th century that has been used for more than 250 years, is the main process in iontophoresis [22,23,24]. In layman’s terms, ER indicates that charged particles promote transport by repelling electrodes of the same polarity, and transport speed is affected by the size and fluidity of the particles themselves. Charged particles are moved by electrophoresis in current-induced transport (EO), whereas uncharged or weakly charged particles are moved by the solvent’s flow, and the direction of transport is consistent with the direction of current flow [25]. In addition to a few intercellular transport channels, skin appendages serve as the primary transport means for iontophoresis. As a result, it benefits from appendage transport, which also includes drug storage and tailored transport.

Large-molecule drugs are typically administered by injection and tend to cause patients to experience pain, skin injury, inflammation, and poor patient compliance [26]. Iontophoresis, according to previous findings, not only allows for transdermal penetration of large molecules but also eliminates the drawbacks associated with injection drug delivery [27]. To compare the effectiveness of enalapril iontophoresis and intradermal injection of enalapril for the treatment of psoriasis in mice, Fukuta et al. performed a preclinical study to assess iontophoresis-loaded enalapril in psoriasis [28]. The results demonstrated that enalapril iontophoresis considerably reduced the signs and symptoms of skin lesions in mice with experimental psoriasis, and this method had a therapeutic effect that was superior to that of enalapril intradermal injection. Dasht Bozorg et al. investigated differences in transdermal absorption of ionized administration in healthy or injured skin when using iontophoresis hydrocortisone to treat psoriasis and eczema [19]. They also compared passive transport and iontophoresis administration for efficacy. The results demonstrated that hydrocortisone could be extensively disseminated to the deeper layers of skin using iontophoresis, with much higher therapeutic benefits than passive hydrocortisone transport, while the drug loading of injured skin was significantly higher than that of healthy skin. However, as systemic corticosteroids are only used under conditions of acute illness, special attention should be paid to the incidence of adverse events when utilizing iontophoresis-loaded hydrocortisone.

Controlled drug release is an advantage of iontophoresis. By regulating the current’s strength and the area of skin that it passes through, the investigator can control the dose of the medicine supplied and prevent major adverse reactions resulting from high drug concentrations [21,29]. Only three products—Ionsys^®^, LidoSite^®^, and Zecuity^®^—have received FDA approval despite significant advances in studies on iontophoresis [30]. Only Zecuity^®^ is currently on the market; the other two products have been discontinued. A major focus of drug research and development is safety. Skin reactions such as itchiness, erythema, and tingling may occur at higher-intensity currents. With time, the risk increases. Therefore, it is crucial to pay attention to the duration and intensity of current use to decrease the likelihood of skin irritation.

#### 3.2.2. Prodrugs

Prodrugs are mostly applied in the second-generation TDDS technique for changing the chemical structure of a drug to increase its penetration effectiveness [31]. The parent drug forms a prodrug through the formation of covalent bonds with substances bearing functional groups such as esters, carbonyls, and amides, improving its pharmacokinetic properties and facilitating its passage through the stratum corneum. The prodrug is inactive by itself, but when it penetrates the stratum corneum barrier, enzymes convert it into a biologically active parent drug that exerts the desired effects [32]. Prodrugs are additionally developed in a quick, reversible, and focused manner for single drugs. Alkylation and polyethylene glycosylation [32] are two techniques for prodrug production that are more frequently employed than chemical derivatization [33]. According to previous reports, the high lipophilicity of alkylated drugs causes them to remain in the skin for long periods while releasing low amounts into the bloodstream. However, the prodrug’s nature makes it less stable than its parent drug; it may hydrolyze when exposed to conditions such as pH and temperature, and increasing the alkyl chain may also decrease the compound’s stability, which could impact the drug’s therapeutic function [34]. Therefore, it is crucial to ensure the main physicochemical properties of the compound (such as its molecular weight) are within a reasonable range when creating a prodrug [35].

In clinical practice, prodrugs are often combined with iontophoresis [36,37]. For instance, Lobo et al. investigated the effectiveness of ion introduction for drug delivery by synthesizing prodrugs (ketoprofen choline chloride (KCC)) with ketoprofen as the parent drug. According to reported findings, employing KCC ion implantation to penetrate human and rat skins is more effective than using iontophoresis alone by factors of 5 and 1.5, respectively [36].

#### 3.2.3. Chemical Penetration Enhancer

The exact process of chemical penetration enhancers is unknown, although studies have proposed two key components. (1) The direct effect of chemical penetration enhancers on the skin temporarily widens the gap between cells by dissolving the phospholipid layer, thereby enhancing the free diffusion of drugs [38]. Chemical penetration enhancers directly interact with drugs to increase their diffusion and distribution coefficients [39]. Numerous substances have been developed as penetration enhancers since the first chemical penetration enhancer was proposed in 1976 [40]. Dipropylene glycol (DIPG), propylene glycol (PG), butanediol (BG), and drugs without permeation-boosting strategies were assessed in a comparative study by Kis et al. [41]. The findings demonstrated that all three chemical penetration enhancers may increase skin permeability compared with the use of none, with PG and BG showing the highest activities.

Currently, ethanol, propylene glycol, isopropyl myristate, oleic acid, penetrating alcohol, dimethyl sulfoxide surfactant, and azone are a few chemical penetration enhancers frequently employed in clinical practice. Indeed, chemical penetration enhancers infiltrate deep skin layers alongside active agents. At large doses, they easily accumulate in the body, which can promote water loss and upset the balance of the body. Additionally, the majority of penetration enhancers are poisonous and irritative by nature, which makes it easy to cause adverse effects and limits their use.

#### 3.2.4. Ultrasound

Ultrasound introduction is a technique that involves using ultrasound waves with frequencies ranging from 20 kHz to 16 MHz to change the lipid bilayer structure of the drug’s stratum corneum, thereby improving drug delivery. Thermal effects and cavitation are among its mechanisms of action [42]. The thermal effect refers to the increase in skin temperature caused by the use of ultrasound, resulting in a higher rate of drug diffusion into the skin; the cavitation mechanism refers to the formation of cavities and bubbles in the stratum corneum as well as the destruction of the stratum corneum’s structure, which is conducive to percutaneous drug transport.

### 3.3. Third-Generation TDDS

The third generation of TDDS is based on transdermal drug delivery that is minimally invasive and destroys the stratum corneum, allowing large-molecule medications and even vaccines to penetrate the skin. Electroporation and microneedling are included in third-generation TDDSs.

#### 3.3.1. Electroporation

Electroporation is a procedure that uses a high voltage (10–1000 V) for a very brief amount of time (less than a few hundred milliseconds) to generate small pores in the skin. The mechanism of electroporation includes exposing a drug solution deposited on the skin to an impulse that creates aqueous pores in the lipid bilayer of the stratum corneum, allowing the drug to reach deeper layers of the skin through the generated pores.

#### 3.3.2. Microneedles

Microneedles feature a cavity for holding drugs with a tiny protrusion in their structure. The first microneedle was successfully created and utilized in the 1990s [43]. The stratum corneum is breached when microneedles are used, allowing the drug to reach the dermis and increasing its effects. Pandey developed a microneedle patch comprising hyaluronic acid for methotrexate dissolution. The latter study discovered that methotrexate microneedles are more clinically efficacious than oral methotrexate at the same dose for skin irritation resembling psoriasis [44].

There are many types of microneedles, each with specific qualities, including solid microneedles, coated microneedles, hollow microneedles, dissolving microneedles, and hydrogel microneedles (Figure 2). Solid microneedles have good mechanical properties, but they can also harm the patient’s skin and make them uncomfortable by inducing skin pores. Coated microneedles are made of a thin layer produced by drugs covering the prepared microneedles. The materials and techniques used to produce coated microneedles are identical to those used for solid microneedles. After the microneedle is introduced into the skin, the coating melts and the drug is released, which requires just one step for the administration process [45]. Coated microneedles also have the advantage of increasing the shelf life of drugs, making them suitable for the administration of strong pharmaceuticals in small doses, e.g., glucocorticoids. However, the ease with which drugs are misplaced highlights their shortcomings. Similar to the design of a micrometer syringe, a hollow microneedle features an internal lumen or pore around 50–70 m in diameter. The pressure released determines how soon the drug can be administered, and the drug is maintained in a hole inside the microneedle [46]. Despite a good drug-carrying capacity, hollow microneedles are fragile; their superior design adds to their cost, and scaling up production is challenging. Soluble microneedles are made from typical biodegradable and biocompatible polymers, including lipids, proteins, and carbohydrates. Their design is different from that of coated microneedles, in which the drug is enclosed in a microneedle matrix, and hollow microneedles, in which the microneedle is dissolved and the drug is administered to the patient in a single, straightforward step after insertion. Although soluble microneedles do not generate medical waste, inadequate skin penetration could result in drug waste [47]. Hydrogel is utilized to produce the latest type of microneedles. During preparation, drugs are either directly injected into microneedles or kept separate in the reservoir, which has a large drug-loading capacity. The good hydrophilicity of hydrogel after application enables it to collect the fluid from the skin tissue; when it does, the volume expands to produce a catheter via which the medication diffuses into the deep tissue of the skin [48]. Hydrogel microneedles can be discarded after use to reduce the generation of medical waste. Coated and dissolving microneedles are currently the most demanded.

The length of the protrusion, typically in the range of 50–200 μm, is correlated with the drug-loading capacity of microneedles [49]. In cases of a too-long protrusion or insufficient mechanical strength, the microneedle is easy to break when inserted into the skin or to penetrate the skin and cause discomfort. When the protrusion is too short, it is difficult for the drug to reach the lesion site, resulting in inadequate drug-loading capacity. Therefore, to achieve adequate drug loading and prevent unneeded side effects, it is advised to utilize microneedles with tiny tip radii in clinical practice. To increase drug penetration into the stratum corneum while eliminating the discomfort and needle phobia associated with large needles, microneedle patches have been developed for transdermal delivery [50]. To treat androgenic alopecia, Wang et al. recently developed the PROTAC-loaded microneedle patch (PROTAC-MNs), which directly distributes drugs to nearby hair follicles through micropores generated by microneedles entering the skin [51]. The composition, mechanism of action, preparation techniques, and examples of current microneedles used in preclinical research to treat skin diseases are briefly summarized in Table 2 for the five different types of microneedles.

### 3.4. Fourth-Generation TDDS

Nanocarriers, besides the aforementioned technologies, have most recently emerged as a separate area for fourth-generation TDDSs. The benefits of nanocarriers can be summarized as follows: They promote transdermal medication absorption, increase the solubility of insoluble drugs, improve targeting, boost drug bioavailability, and reduce unfavorable side effects. Liposomes, transferosomes, ethosomes, niosomes, cubosomes, solid lipid nanoparticles (SLNs), nanostructured lipid carriers (NLCs), nanoemulsions, microemulsions, polymer nanoparticles, polymer micelles, dendrimers, etc., are examples of common nanocarriers (Figure 3). The benefits and drawbacks of nanocarriers will be briefly discussed in the following sections, along with a list of current studies that have addressed these issues (Table 3).

#### 3.4.1. Lipid Vesicular Carriers

Liposomes were the first lipid-based carrier system to be examined for usage. Liposomes, which are composed of phospholipids and cholesterol as a membrane stabilizer, are generally regarded as safe carriers and are commonly utilized for lipophilic and hydrophilic medicines. Liposomes are solid lipid nuclei and phospholipid-based, closed spherical vesicles. Psoriasis is managed with cyclosporine (CYC), a calcineurin inhibitor. Previous findings have demonstrated that oral administration of CYC is ineffective, tends to cause rebound after withdrawal, and may have very negative side effects. Walunj et al. evaluated the therapeutic potential of a cationic liposome carrier gel containing CYC in a mouse model of imiquimod-induced psoriasis [94]. According to these findings, liposomes improved CYC’s affinity for skin membranes, decreased the levels of key psoriatic cytokines such as TNF, IL-17, and IL-22, and considerably lessened skin lesions in psoriatic mice. Liposomes function as medication localizers by operating on the skin’s outer layer while being larger, stiffer, less stable, and less often able to penetrate deeper into the skin [95,96,97]. Another issue is that drug leakage from liposomes is brought on by phospholipid hydrolysis.

To circumvent the limitations of ordinary liposomes, Cevc and Blume developed transferosomes in 1992, which were registered by IDEA AG in Germany [98]. By incorporating edge-active compounds, structural and chemical alterations were made on the basis of conventional liposomes [99]. Transferosomes have tremendous elasticity and deformability, allowing them to pass through tiny pores much smaller than their own, allowing them to get beyond the SC barrier and into deeper layers of the skin. Parkash et al. created transferosomes using rotary evaporation to increase tacrolimus transdermal permeability and investigated medication penetration and anti-psoriasis efficacy. The results showed that tacrolimus transfersomes had better permeability, which was consistent with the increased anti-psoriasis activity in in vivo and in vitro studies. However, as the permeation of the transferosomes is dependent on the water gradient, they are challenging to transport under occlusive conditions and also challenging to load with hydrophobic drugs.

Ethosomes are another novel lipid carrier consisting of phosphatidylcholine, cholesterol, ethanol, and water that can penetrate deep into the skin and body circulation. The main advantage of ethosomes is that they can be loaded with both hydrophilic and lipophilic drugs while enabling effective drug delivery under both occlusive and non-occlusive conditions. Compared with traditional liposomes, the presence of cholesterol and ethanol increases the elasticity of the carrier while also increasing its stability. Although the short-term effects of ethosomes have been addressed in the literature, there is still a lack of long-term efficacy evaluation.

Niosomes constitute a useful carrier for TDDS therapy for skin conditions since non-ionic surfactants are reasonably inexpensive and show great biocompatibility with the epidermis [100]. Niosomes are considered to have two roles for increasing the efficiency of drug penetration: (1) Extending the amount of time that drugs are retained in the stratum corneum, increasing the concentration gradient caused by drug accumulation, and enhancing passive transport; and (2) acting directly on the stratum corneum, loosening its structure and decreasing resistance to drug penetration [101]. Around the world, 2 to 3% of people have psoriasis [102]. Although celastrol has demonstrated specific benefits in the treatment of psoriasis, its low molecular weight, water solubility, and permeability diminish its effectiveness. To address these problems, a unique celastrol formulation, celastrol–niosome hydrogel, was developed by Meng et al. [103]. Transdermal penetration of a substance was significantly increased compared with the original medication. Additionally, it considerably decreased the frequency of adverse effects and skin lesions in psoriatic mice. Desoximetasone niosomes with a hydrogel formulation were developed by Shah et al. to treat psoriasis [3]. The latter study demonstrated that the hydrogel formulation of dexamethasone niosomes resulted in fewer systemic adverse reactions and caused less medication to enter the systemic circulation than the reference gel preparation. The main limitations of non-ionic surfactant liposomes are their relatively high cost and extremely stringent safety-related criteria, which prevent their easy commercial availability.

#### 3.4.2. Lipid Nanoparticles

SLNs, a member of the first-generation lipid nanocarriers, were first introduced in 1991 and used to replace oil in emulsions with water [104]. Surfactants and a large lipid nucleus make up most of the SLNs. The active ingredient of the surfactant, which is intended to increase SLN stability, can either be maintained in the lipid nucleus or bonded to the surface of SLNs. The solid lipid nucleus maintains its solid state when applied transdermally at ambient temperature. Thanks to their microscopic size, high stability, and ability to completely interact with lipid components in epidermal cells, SLNs combine the advantages of polymeric nanoparticles and liposomes. Additionally, they have good biocompatibility. The key factors affecting the widespread use of these carriers include controlled drug release, low toxicity, biodegradability, and scalability. That SLNs produce a film on the skin’s surface to exert a good occlusion effect, which lowers epidermal water loss and is especially useful for skin injuries, is another advantageous property of SLNs [105]. Due to its anti-inflammatory and anti-pruritic qualities, mometasone furoate (MF) is recommended for chronic inflammatory skin diseases. Despite the existence of MF lotion and gel treatments, their use only slightly reduces irritation. Madan et al. assessed MF SLN gels, which increased both the rate of skin deposition and the duration of drug activity in comparison with commercially available gels. Clinically effective SLN gels offer a sustained release period of up to 8 h and a 2.67-times higher skin deposition than commercially available gels [106]. There are still issues with drug leakage and a constrained capacity for drug loading in SLNs, with the drug’s poor solubility having a huge effect. In general, adding an extra liposome can increase stability and the encapsulation effect [42]. NLCs have been created to improve drug delivery capabilities and address the shortcomings of SLNs.

NLCs constitute one type of lipid nanocarrier that belongs to the second generation, with particles of 150–300 nm in size [107]. Because NLCs’ lipids are split into two groups—solid lipids and liquid lipids (oil), for example—this differs significantly from SLNs in this regard [108]. Besides lowering the drug’s water content, increasing the melting point of the solid matrix, preventing drug leakage, and enhancing drug stability, adding oil also enhances the drug’s loading capacity. The specific advantages of applying NLCs by percutaneous methods are: (1) The lipid component adheres to the skin surface to generate a lipid film with a good occlusive effect and suitability for inflamed or impaired skin; (2) high safety, i.e., regarded as the safest delivery tool; (3) to prevent drug degradation, the active pharmaceutical ingredient is encased in a lipid matrix; additional requirements include (4) biocompatibility, (5) photostability, and (6) good lubricity [109]. Mycosis can be treated with luliconazole, and Mahmoo et al. generated NLC hydrogels to increase penetration efficiency and prolong sustained release action [110]. In vitro pharmacokinetic tests revealed that drug concentrations in the epidermis and dermis are much higher than those of commercially available ointment products. In vitro drug release experiments have revealed drug release times of up to 42 h. The main drawbacks of NLCs are their vulnerability to phase changes and the paucity of long-term stability data.

#### 3.4.3. Emulsion Based Carriers

Nanoemulsions and microemulsions are two distinct types of surface-active agent-based nanocarriers, both of which include a water phase, an oil phase, surfactants, and co-surfactants. They are categorized as completely different prescriptions despite having striking structural similarities and radically varying particle sizes, preparation techniques, and thermodynamic stability [111]. Nanoemulsions are kinetically stable systems that require a lot of energy to manufacture, with typical particle sizes ranging between 50 and 200 nm [112]. Besides the advantages of reduced irritancy and regulated medication release, the pro-permeation impact of nanoemulsions is principally produced by changing the spatial organization of the stratum corneum. Although this action is essential in the topical management of dermatological conditions, their low viscosity is a disadvantage. By creating co-delivery of imiquimod and curcumin through a nanoemulgel, MS Algahtani et al. assessed whether combining imiquimod and curcumin has a synergistic impact. According to their results, imiquimod nanoemulsions considerably outperformed imiquimod polymer gel in treating psoriatic skin lesions [113]. In psoriatic mice, imiquimod and curcumin nanoemulsions could overtly relieve the signs of skin lesions. 8-methoxypsoralen (8-MOP), a drug, has severely serious adverse effects after oral treatment, including phototoxicity, hepatocellular cancer, tachycardia, and depression. Due to its high molecular weight, which prevents the drug from fully entering the epidermal layer, its use is similarly limited for topical treatment. Thas et al. could improve the penetration of psoralen and overcome the low viscosity of nanoemulsions by using clove oil and anise oil to create 8-MOP hydrogel nanoemulsions [114].

Particles as small as 100 nm in diameter spontaneously generate a dynamic, thermodynamically stable system termed a microemulsion. For the following reasons, microemulsions specifically boost medicine penetration: Drug diffusion through the dermal appendage route and building up in hair follicles are two ways to increase the solubility of insoluble drugs and increase the diffusion coefficient [115]. Other methods include utilizing the oil phase and emulsifiers as chemical permeation promoters and direct contact with the skin to change the lipid bilayer structure. To assess how the microemulsion affects the cutaneous distribution of hydrophobic drugs, Zhang et al. tested the antibacterial efficacy of the microemulsion using the hydrophobic drug clotrimazole (CLOT) [116]. Human skin penetration tests have demonstrated that adding more water to microemulsions significantly increases the effectiveness of CLOT delivery to the dermis. According to in vitro solubility assays, CLOT exhibited the lowest solubility in aqueous PBS (5 × 10^−5^ mg/mL) and the maximum solubility in the surfactant (73.8 mg/mL). With naftifine, fungal infections may be treated. Erdal et al. developed a naftifine microemulsion using oleic acid (oil phase), Kolliphor EL or Kolliphor RH40, Transcutol, and water [117]. According to in vitro transdermal studies, the naftifine microemulsion significantly increased the efficiency of transdermal penetration compared with other commercially available dosage forms, and an analysis by infrared spectroscopy demonstrated the microemulsion increased the lipid bilayer’s mobility. As innovative drug delivery systems, microemulsions and nanoemulsions improve the solubility and bioavailability of pharmaceuticals and can be employed for insoluble drugs. Low-viscosity liquid formulations, including microemulsions and nanoemulsions, are difficult to administer. These formulations also have higher concentrations of surfactants, which may harm the skin. To solve this issue, investigators have tried merging nanoemulsions and microemulsions; however, this study is still in the exploratory stage, and its viability needs to be confirmed.

#### 3.4.4. Polymeric Nanoparticles

Recently, polymeric carriers have attracted increasing attention, notably polymeric nanoparticles and polymeric micelles, for the treatment of dermatological disorders. Polymeric nanoparticles primarily alter the structure of the stratum corneum to enhance penetration efficiency. They can be produced with synthetic or natural polymers. Polymeric nanoparticles have the advantages of high drug encapsulation rates, prevention of drug degradation, and controlled drug release regulation. Hussain et al. developed polymer nanoparticles that were concurrently loaded with curcumin and quercetin to treat burn wounds [118]. The effects of encapsulation on stability, permeability, and wound healing were then investigated and evaluated. The results demonstrated the drug’s effective loading and encapsulation rates as well as its advantageous effects on wound healing. Balzus et al. found that ethylcellulose, Eudragit^®^RS, loaded with dexamethasone, may control drug release and penetration in the treatment of inflammatory skin diseases [119]. To effectively treat psoriasis and atopic dermatitis, Caon et al. found that poly-caprolactone-loaded curcumin significantly increased drug deposition on the skin surface and reduced drug waste [120]. Polymer nanoparticles have large particle sizes. The drawback of polymer nanoparticles is that they frequently have particle sizes of approximately 1000 nm, which makes it challenging for these products to pass through intercellular and intracellular routes and enter the skin [121]. They are most effective in disorders affecting hair follicles because they accumulate in the follicle and follow the accessory apparatus pathway to carry the drug to the dermis. Polymer nanoparticles can support pharmaceutical penetration in the medical context in addition to other physical techniques such as iontophoresis, microneedles, ultrasound, etc.

Amphiphilic polymers can self-assemble into polymer micelles, which are spherical structures with diameters below 100 nm [122]. Their center can only dissolve lipophilic and some hydrophilic drugs, unlike their outer shell, which may dissolve hydrophilic drugs [123]. Because systemic drugs are prone to structural changes due to the effects of the internal environment, polymer micelles are widely utilized as external drug carriers [124]. The colloidal system can considerably increase drug solubility with good stability, which makes it more effective in penetrating skin barriers. Since it can also target drug delivery to the site of action, this system is commonly utilized to treat skin conditions. Oral ciclosporin A (CSA) can be used to treat severe psoriasis but has side effects on the liver and kidneys. Lapteva et al. developed a topical treatment containing CSA polymeric micelles, which was tested in vitro on pig ear skin to gauge its selective penetration [125]. The outcomes demonstrated that, in comparison with standard formulations, only a very small amount of the drug was absorbed into the body, resulting in reduced side effects, which may be related to the intercellular movement of polymeric micelles. Kahraman et al. developed an enhanced polymeric micelle carrier of benzoyl peroxide (BPO) for acne treatment that was significantly more effective than commercial gels [126].

Dendrimers are three-dimensional, highly branching macromolecular structures that can be employed to encapsulate hydrophobic and hydrophilic medicines. According to reports, dendrimers exhibit ligand-targeting characteristics, cause minimal skin irritation, and increase medication deposition and penetration. As a result, it is frequently utilized in the treatment of melanoma and squamous cell carcinoma. Dendrimers such as polyamide amide (PAMAM) and polypropylene amide (PPI) are widely employed in medication delivery. PAMAM dendrimers loaded with indomethacin had a considerably higher drug concentration in the blood of Wistar rats than the drug suspension. Dendrimers are unsuitable for delivering hydrophilic medicines because the materials and costs are prohibitively expensive. Controlling drug release is a challenge for dendrimers at the moment, and there is little toxicological data to prove their safety.

#### 3.4.5. Nanocrystals

Arepitant nanocrystals were developed in the 1990s and legally commercialized in 2000. A nanocrystal is a type of pure drug particle with a high drug-loading capacity. The specific surface area of the drug particle increases with decreasing particle size, enabling full skin contact. The drug’s saturation solubility and penetration efficacy both increase concurrently. Another factor contributing to the high penetration effectiveness of nanocrystals is that they pass through appendages, especially hair follicles, and the drug accumulates as a result, enhancing the concentration gradient. After adding a small amount of surfactant to the formulation as an excipient to promote stability, drug particles are bound to the surfactant’s surface, which can help prevent drug aggregation [127]. The following characteristics highlight the advantages of nanocrystals [128]: The factors that make a drug carrier suitable for mass production include: (1) Greater saturation solubility and dissolution rate compared with large-size drug carriers, promoting passive drug transport; (2) enhanced drug loading and dosage reduction; (3) high stability; (4) good safety due to low excipient content in the formulation; and (5) high stability. Pireddu et al. developed diclofenac nanocrystals by wet medium grinding technology, using poloxamer 188 as a stabilizer and diethylene glycol monomer as a penetration booster [129]. The effectiveness of the transdermal transport of diclofenac nanocrystals was evaluated in vitro using Franz diffusion cells and pig skin. The results demonstrated synergistic effects of Poloxam 188 and the penetration enhancer for diethylene glycol monoethyl ether on the transdermal penetration of diclofenac. Apste has low lipophilicity, which makes it challenging to penetrate the stratum corneum. Parmar et al. developed Apster nanocrystals that are twice as soluble as Apster [130]. It was demonstrated that Apster nanocrystals possessed stratum corneum and dermis permeabilities that were 2.6 and 3.2 times higher than those of Apst crystal, respectively, in a Franz diffusion cell utilized to research skin pharmacokinetics. Olga Pelikh et al. found that the smaller the nanocrystals, the better the drug penetration, but shorter drug residence time on the skin was caused by improved mobility, and they can typically be produced as nanocrystalline oleogel and cream formulations [131].

Combining two or more delivery vehicles, each of which has advantages and drawbacks, might enhance drug penetration. Table 2 briefly summarizes several types of nanocarriers in terms of matrix material, permeation mode, and preparation technique and exemplifies preclinical studies on nanocarriers for the treatment of dermatological diseases in recent years.

## 4. Challenges and Outlook

Treatment with injectable medications is non-specific and can induce significant systemic toxicity based on traditional oral doses. The TDDS is currently becoming a potent solution for treating skin conditions. Governmental authorities are increasingly and strictly supervising the development and commercialization of new TDDS products, besides encouraging TDDS clinical trials. To assess the development trends of TDDS and other treatments, we searched clinical trials of drugs registered for the treatment of dermatological diseases in recent years on http://ClinicalTrials.gov (accessed on 1 March 2023). We also compared the percentage of clinical trials for TDDS at various stages. The development trend is roughly consistent with the overall drug use for dermatological diseases, as shown in Figure 4, which details 1108 TDDS clinical trials for dermatological diseases that have been registered on http://ClinicalTrials.gov (accessed on 1 March 2023) over the past 10 years, of which 672 (60.6%) were successfully completed. Except for years 2020–2022, when the COVID-19 pandemic caused a reduction in clinical trials, the overall trend was upward. Phase II and phase III clinical trials saw the most successfully registered TDDS clinical trials (34% and 20%, respectively), whereas phase IV clinical trials had comparatively few registrations (13%). Although preclinical studies have shown potential benefits for the TDDS, many products are still in the development stage, and further research is required to confirm the efficacy and safety of these treatments. Some TDDSs that have been listed are shown in Table 4.

### 4.1. Safety

Safety is given primary attention when developing new drugs. Adverse TDDS-related effects are common in daily life. TDDSs can still have harmful local effects such as skin atrophy, cytotoxicity, and phototoxicity, even though they can minimize systemic side effects [4,132]. For instance, whereas NLC is considered a safe carrier, this formulation also includes a small amount of a surfactant that may be harmful [133]. Skin lesions formed after microneedle administration are quite small, but because of their ability to penetrate the epidermis, they may transmit dangerous bacteria and cause unfavorable side effects such as inflammation. Despite its high rate of encapsulation and ease of production, ethanol can irritate the skin at high concentrations [134]. To maximize the drug’s effectiveness, investigators have examined higher doses, which only resulted in multiple adverse effects. Since the TDDS has no remarkable local effects, some drugs may be absorbed transdermally and exert systemic effects. When using a TDDS, especially for damaged skin, the dose and frequency of drug administration should be carefully controlled to minimize absorption toxicity [135,136,137]. Although several gadgets may help the drug penetrate into the skin, they have not yet been considered for a defined technique to verify its dependability [138]. The government should enhance its quality control since the development of the TDDS necessitates an accurate assessment of the associated detrimental effects.

### 4.2. Difficulties in Large-Scale Production

High costs and unresolved major difficulties are challenges in progressing from laboratory design to large-scale industrial manufacturing [139]. The manufacturing environment is more complex in the workshop than in the laboratory since the latter setting is larger and more difficult to maintain in terms of temperature, humidity, aseptic operation, and other test conditions. If not treated carefully, there is a possibility of infection. Repeatability between batches must also be considered in industrial production to avoid inconsistent pharmacological effects. The creation of the TDDS is more challenging than the development of injectable and oral drug delivery tools. Manufacturers desiring to produce microneedle patches on a large scale, for instance, have challenges because of their complex structures, which elevate development costs and make it challenging to maintain high standards for quality [140]. Additionally, the TDDS is more sensitive to pharmacokinetic and thermodynamic effects. To prevent major impacts of medication degradation on efficacy, it should be assured that appropriate environmental parameters (light, temperature, and humidity) are met during production, packaging, storage, and ultimate use. It is crucial for researchers to consider future strategies for achieving industrial production of the TDDS on a large scale.

### 4.3. Lack of Standards for Bioequivalence Evaluation

The unpredictable nature of bioequivalence testing procedures is one of the factors hindering the development of the TDDS. Bioequivalence is a method for evaluating the key pharmacokinetic characteristics of reference formulations for generic drugs. The FDA and European Medicines Agency (EMA) have provided no precise guidelines so far, and there is a great deal of interest in studying the bioavailability of the TDDS [141]. Due to the unique delivery method, diverse targets, and differences between animal and human skins, the current methods for determining the bioequivalence of TDDS are not standardized, with each having benefits and drawbacks. The evaluation methods mostly used by regulatory agencies for the majority of generic drugs applied with a TDDS are clinical endpoint studies and pharmacodynamic investigations [142].

Clinical endpoint studies, the most popular bioequivalence evaluation method used for all types of drugs, are the “gold standard” for assessing the bioequivalence of TDDS generics. Randomized controlled clinical trials, on the other hand, are typically expensive and time-consuming, demand a large number of participants, and are challenging for ethical considerations.

Studies on pharmacodynamics are based on the temporal link between a drug’s pharmacological action and its effects [143]. The sole pharmacodynamic method with FDA approval is the vasoconstriction test, which is currently broadly applied as an alternative method for clinical endpoint studies in multiple nations but is typically only applicable to corticosteroids. Its basic tenet is the use of topical corticosteroids, which cause overt paleness and skin vascular constriction, both of which are strongly associated with the clinical efficacy of the drug. Through ocular assessment, chromaticity measurement, and digital image analysis, the degree of pallor may be gauged.

Several supplementary therapies are also available, including the following. (1) In vivo pharmacokinetic research. Even though the TDDS is typically suitable for oral drugs, it does not penetrate systemic blood circulation, which hinders its frequent application; however, when it enters circulation, it can be used to identify adverse systemic effects. (2) The tape adhesion method. This involves applying the TDDS topically, covering a portion of the stratum corneum with adhesive tape, removing the drug from the tape with an appropriate solvent, and measuring drug concentration using techniques such as high-performance liquid chromatography (HPLC) and spectrometry. The tape adhesion method was formally abandoned in 2002 as it is only applicable to the bioequivalence evaluation of drugs acting in the stratum corneum and cannot adequately assess the effectiveness of drugs whose targets are in the dermis or subcutaneous tissues [144]. This technique assesses drug concentration using a hollow probe and a microdialysis pump. The continued development and validation of the method are constrained since variations in the skin site where the probe is placed would significantly affect the results, and it cannot be used for medications slowly penetrating the skin [145]. The bioequivalence of the TDDS has yet to be determined using a gold standard. Based on physicochemical properties and the sites of pharmacological action, appropriate strategies for application should be selected.

## 5. Conclusions

The TDDS attracts increasing attention because it addresses problems with oral pharmaceutical bioavailability, the inconvenience of using injectable drugs, pain, and uncontrolled drug release. Additionally, the TDDS improves penetration efficiency while also increasing targeting, stability, and effectiveness compared with conventional topical medicines. Because most related research is still in the preclinical stage, further clinical trials are needed to evaluate the effectiveness and safety of the TDDS. The TDDS has high potential and substantial commercial value for dermatological treatments.

## Figures and Tables

**Figure 1 pharmaceutics-15-02165-f001:**
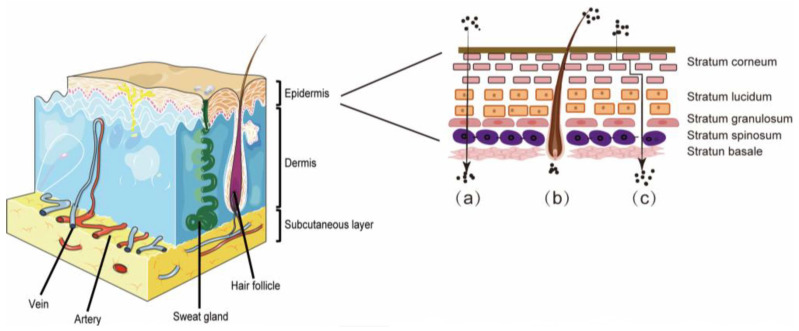
This diagram depicts the structure of the skin and the three modes of percutaneous penetration of the TDDS: (a) intercellular lipid route; (b) appendageal pathway; and (c) transcellular pathway.

**Figure 2 pharmaceutics-15-02165-f002:**
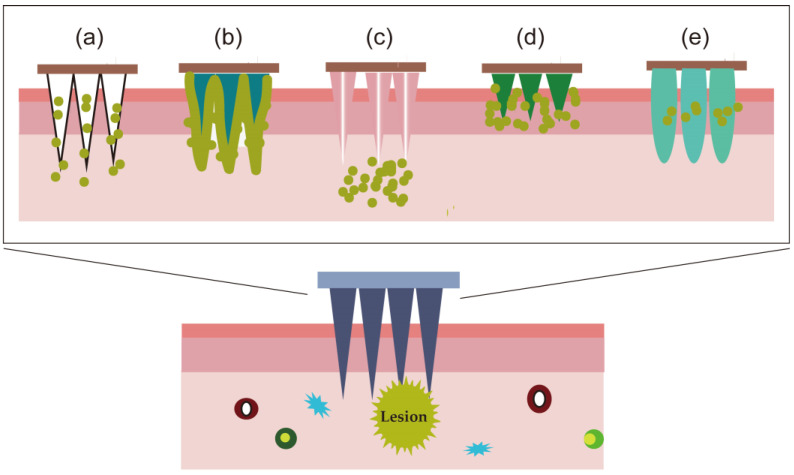
Characteristics of different types of microneedles for percutaneous penetration. (**a**) Solid microneedles; (**b**) coated microneedles; (**c**) hollow microneedles; (**d**) dissolving microneedles and (**e**) hydrogel microneedles.

**Figure 3 pharmaceutics-15-02165-f003:**
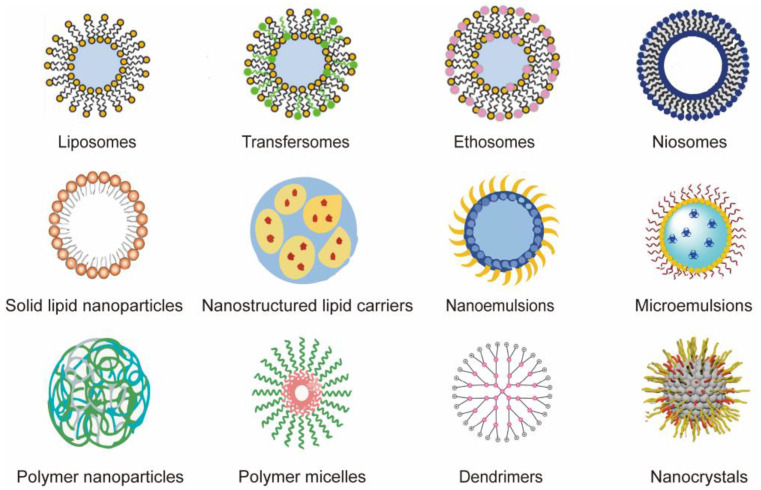
Different types of nanocarriers used as controlled delivery vehicles for treating dermatological diseases.

**Figure 4 pharmaceutics-15-02165-f004:**
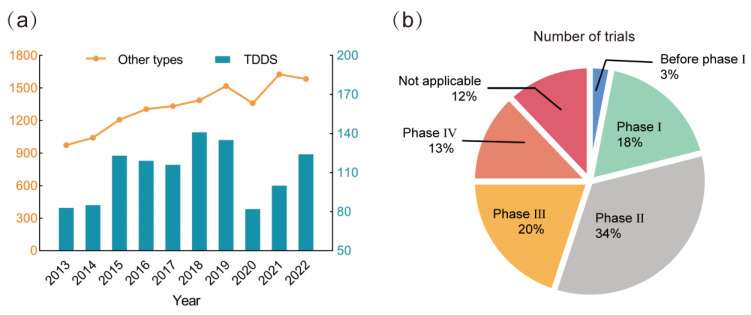
Clinical studies of TDDS for skin diseases conducted in the last decade, derived from https://clinicaltrials.gov/ (accessed on 1 March 2023). (**a**) Differences in trends between TDDS and other treatment methods. (**b**) Percentage of clinical trials in different trial phases.

**Table 1 pharmaceutics-15-02165-t001:** Second generation transdermal preparations.

Transdermal Technique	Mechanism	Advantage	Disadvantage	Drug
Ultrasound	Heat effect and cavitation effect	Delivering many different types of drugs	Higher precision instrument requirements; impacted by ultrasonic frequency, intensity, and mode	Ketoprofen [18]
Iontophoresis	Electro-rejection and electro-osmosis	Realization of macromolecule transdermal penetration	Higher demands on instrumentation; complex to use	Hydrocortisone [19]
Prodrugs	By attaching the inactive ingredient to the medicine, the parent drug becomes more hydrophobic than the active form	Targeted	Designed specifically for a particular drug	Stavudine [20]
Chemical penetration enhancer	Direct interaction with the stratum corneum or the drug to improve drug penetration efficiency	Small-molecule drug transdermal penetration	Toxic; may cause skin irritation when used in high concentrations	

**Table 2 pharmaceutics-15-02165-t002:** Transdermal drug delivery studies based on microneedle carriers.

Category	Materials of the Substrate	Mechanism	Preparation Methods	Drug	Treatable Diseases
Solid microneedles	Silicon, titanium, stainless steel, and other polymer materials insoluble in water	Does not contain drugs and leaves micropores in the skin during use. The active drug components penetrate the skin through these micropores, belonging to passive transport	Etching process, mechanical cutting	Acyclovir [52]	Herpes
Coated microneedles	Metal or polymer materials	After insertion into the skin, the drug coating dissolves from microneedles and quickly enters the tissue for one-step administration	Dip coating method, gas jet drying method, and spraying method	Bleomycin [53]	Plantar wart
Hollow microneedles	Polymer materials that are insoluble in water, including silicon, glass, stainless steel, etc.	The drug penetrates into the skin under pressure, acting like a microsyringe	Lithography technology	Synthetic [54] mRNA [55]	Skin diseases
				Vaccinum [56]	
				and tofacitinib citrate [57]	Psoriasis, alopecia areata, and vitiligo
Soluble microneedles	Polymer materials with degradability and biocompatibility (e.g., maltose, carboxymethyl cellulose, etc.)	After insertion into the skin, the needle tip matrix remains in the skin while the drug is released, requiring only a one-step application	Hollow method, centrifugal method, fusion method, and casting method	Methotrexate [58]	Psoriasis
				Cisplatin [59]	Superficial tumors
				Amphotericin B [60]	Mycosis
				Nanoemulsion [61]	
				Cyclosporin A [62]	Psoriasis
				Tofacitinib citrate [57]	Psoriasis
				Triamcinolone Acetonide [63]	Psoriasis
Hydrogel microneedles	Expandable hyperlinked polymer	By absorbing tissue fluid and expanding in the skin, porous microducts are formed through which drugs can be diffused into the skin microcirculation	Vacuum method, centrifugal method after crosslinking, and freeze-drying method	Sorbitol [64]	
				Insulin [65]	Diabetic wound

**Table 3 pharmaceutics-15-02165-t003:** Transdermal drug delivery based on nanocarriers.

Category	Penetration Method	Advantages	Limiations	Medication	Treatable Diseases
Liposomes	Intercellular pathway	low toxicity; biocompatibility and biodegradability; simple production process	Low stability; large volume and lack of elasticity	Adapalene and benzoyl peroxide combination [66]	Acne
Transfersomes	Intercellular pathway	Highly elastic; deformable	Hydrophobic drug loading is challenging; loading hydrophobic drugs poses challenges	Dexamethasone [67]	Skin disease
				Triamcinolone [68]	Skin disease
Ethosomes	Intercellular pathway	Suitable for hydrophilic and lipophilic drugs; can be used under both blocked and non-blocked conditions	Long-term impact still needs to be evaluated	Paclitaxel [69]	Skin cancer
				Tacrolimus [70]	Atopic dermatitis
Niosomes	Intercellular pathway	High stability		Tripterygium wilfordii [71]	Psoriasis
Cubosomes	Intercellular pathway	Good adhesion performance; thermodynamic stability	Insufficient carrier materials; research on the lack of in vitro transdermal performance	Cinnamaldehyde [72]	
Binary ethosomes	Intercellular pathway			Vismodegib [73]	Skin cancer
SLNs	Accessory pathway	High stability; low toxicity; good flexibility	High moisture content; low drug loading; tends to gel	Mirtazapine [74]	Itch
				Fluconazole [75]	Pityriasis rosea
				Tacrolimus [76]	Atopic dermatitis
				Combination of isotretinoin and α-tocopherol [77]	Acne
NLCs	Accessory pathway	High drug-loading capacity; high stability; high biodegradability and biocompatibility; suitable for large-scale production	Tend to gel; lack of long-term stability data	Itraconazole [78]	Fungal infection
				Mometasone furoate [79]	Psoriasis
				Adapalene combined with vitamin C [80]	Acne
				Curcumin [81]	Chronic inflammatory diseases, psoriasis, acne
Nanoemulsions	Accessory pathway	Improve solubility; enhanced permeability	Irritability; low stability; low viscosity	Coumestrol [82]	Herpes
				Methotrexate [83]	Psoriasis
Microemulsions	Accessory pathway	Mass production; thermodynamic stability	Toxicity	Cyclosporin [84]	Psoriasis
				Retinol palmitate [85]	Acne, skin aging, psoriasis
				Ivermectin [86]	Parasite infestation
Polymer nanoparticles	Accessory pathway	High stability; targeting	Difficulties in large-scale production	Betamethasone Valerate [44]	Atopic dermatitis
				Tacrolimus [87]	Atopic dermatitis
				Methotrexate [88]	Inflammatory diseases
Polymer micelles	Intercellular pathway	Accurate release	Limited to lipophilic drugs; low drug-loading capacity	Imiquimod [89]	Basal cell carcinoma
				Adapalene [90]	Acne
				Benzoyl Peroxide	Acne
				combination of Indomethacin and Resveratrol [91]	Skin cancer
Dendrimers	Intercellular pathway	Increase the solubility of high lipophilic drugs; Targeting	Not suitable for hydrophilic drugs; cytotoxicity; high cost	8-methoxypsoralene [54]	
Nanocrystals	Accessory pathway	High solubility; high drug-loading capacity; scalable production	Difficulty in optimizing size and dosage	Curcumin [92]	
				Miconazole nitrate [93]	Fungal skin disease

**Table 4 pharmaceutics-15-02165-t004:** TDDS already on the market.

Trade Name	Formulations	Indication	Time to Market
Ztlido	Patch	Relieve neuropathic pain associated with herpes zoster	February 2018
Naftin	Gel	Foot moss	June 2013
Aczone	Gel	Acne	July 2015
Impoyz	Cream	plaque psoriasis	November 2017
Vectical	Ointment	plaque psoriasis	January 2009
Finacea	Foam agent	Lupus erythematosus pustule	July 2015
Altreno	lotion	Acne vulgaris	November 2018

## Data Availability

The data are available from the corresponding author on request.

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
