# Peer review of "Advance and Challenges in the Treatment of Skin Diseases with the Transdermal Drug Delivery System"

_pharmaceutics, 2023, doi:10.3390/pharmaceutics15082165_

Round 1
Reviewer 1 Report
Lot of literature already published on such topic; I don’t find novelty in this review article.
In my opinion this topic is not novel, the article doses not score high for publication in this high impact journal.
Manuscript should be significantly revised considering including literature on Ethosomes, Transfersomes, Cubosomes, Nanofibers, dendrimers, transethosomes, binary ethosomes are missing.
Figures demonstrating the major penetration pathway of studied nanoparticles or nanovesicles and droplets should be included in the manuscript.
Advantages and disadvantages of each transdermal system should be included in the manuscript.
Skin cancer is completely missing in this article, lot of literature are available.
Author Response
Reviewer #1
Lot of literature alredady published on such topic;I don’t find novelty in this review article.In my opinion this topic is not novel, the article doses not score high for publication in this high impact journal.
Response: Thank you for the comments. In the manuscript, we reviewed the development history of TDDS, and the content involved in this part inevitably quoted some new references. Based on your suggestion, we have added citations to the recently published literature.
1.Manuscript should be significantly revised considering including on Ethosomes, Transfersomes, Cubosomes, Nanofibers, Denderimers, Transethosomes, Binary ethosomes are missing.
Response: Thank you for your valuable feedback. Due to the abundance of nanocarriers, we initially only considered introducing some commonly used types and neglected the description of Ethosomes, Transfersomes, Cubosomes, Nanofibers, Denderimers, Transethosomes, Binary ethosomes. We have adopted your suggestion and added research on these carriers.Please see section 3.4. and Table 3.
2.Figures demostrating the major penetration pathway of studied nanoparticles or nanovesicles and droplets should be included in the manuscript.
Response: Thank you very much for your advice. The penetration pathways of nanocarriers mainly include intracellular pathway, intercellular pathway and skin accessory pathway, which have been described in Figure 1, so the part of nanocarriers is not described again. The potential drug penetration enhancement mechanism of nanocarriers is mainly related to the types of nanocarriers, including their compositions, such as polymers, lipids or surfactants, etc. We have added descriptions of various transdermal penetration pathways of nanocarriers in the manuscript and in Table 3.
3.Advantages and disadvantages of each transdermal system should be included in the manuscript.
Response: Thank you for your suggestion. We have made corresponding modifications to our manuscript. Please see the relevant section.
4.Skin cancer is completely missing in this article, lot of literature are available.
Response: Thank you for your suggestion. Indeed, the incidence rate of skin cancer is very high, and we apologize for ignoring the research related to skin cancer. We have added some studies on the treatment of skin cancer with transdermal preparations. Please see page 13 and Table 4.

Reviewer 2 Report
The manuscript by Cheng and collaborators describes with detail the development and progress on TDDS. It is well written and organized. In my opinion more focus could be given to TDDS for diseased skin, as SC is usually damaged and the relevance of TDDS is questionable. Some discussion is present, but I recommend the authors to improve it.
Some few recomendations
a) in the keywords should be in alphabetical order, and preferably do not repeat words in the title; suggestion replace development by advances
b) please indicate in figure 1B the stratum corneum, given its relevance should be represented
Author Response
Reviewer #2:
The manuscript by Cheng and collaborators describers with detail the development and progress on TDDS. It is well written and organized. In my opinion more focus could be given to TDDS for diseased skin, as SC is usually damaged and the relevance of TDDS is questionable.
Response: Thank you for your positive comments and suggestions on our manuscript. We have tried our best to revised the manuscript.
1.In the keywords should be in alphabetical order,and preferably do not repeat words in the title; suggestion replace development by advances.
Response: Thank you for your suggestion. We have revised the keywords according to your suggestions. Please see Page 2.
2.Please indicate in figure 1B the stratum corneum,given its relevance should be represent.
Response: Thank you for your suggestion. We have revised Figure 1B.

Reviewer 3 Report
This review “Developments and challenges in the treatment of skin diseases with the transdermal drug delivery system” is interesting. The authors have provided an overview of various TDDS used in skin diseases. I have a few suggestions to improve this manuscript.
Comments
1. In Fourth-generation TDDS (nanocarriers), authors should include a brief statement of their mechanism of transport.
2. The title of this review mentioned “Challenges”, however, the authors did not discuss the challenges in the text. I wish the authors emphasize this in each section and highlight the possible options to overcome those challenges, which will help the readers.
3. Another suggestion is to include the clinical outcome of the formulations commercially available.
4. References are inconsistent.
Minor grammar corrections required.
Author Response
Reviewer #3:
1.This review “Developments and challenges in the treatment of skin diseases with the transdermal drug delivery system” is interesting. The authors have provided an overview of various TDDS used in skin disease. In Fourth-generation TDDS (nanocarriers), authors should include a brief statement of their mechanism of transport.
Response: Thank you for your positive comments and suggestions. We have tried our best to revise our article. The transdermal penetration of nanocarriers includes intracellular pathway, intercellular pathway and appendicular pathway. Generally, complete nanocarriers cannot directly pass through the stratum corneum, but can accumulate in the stratum corneum to provide a drug reservoir and improve the efficiency of drug transdermal penetration. It can also penetrate the skin through appendages such as hair follicles, especially lipid nanocarriers.
2.The title of this review mentioned “challenges”,however, the authors did not discuss the challenges in the text. I wish the authors emphasize options to overcomes those challenges, which will help the readers.
Response: Thank you for the comments. Our manuscript presented the challenges of TDDS at this stage in section 4 (Challenges and Outlook).
3.Another suggestion is to include the clinical outcome of the formulations commercially available.
Response: Thank you for your suggestion. We have made revisions to our manuscript. Please see Table 4.
4.References are inconsistent.
Response: Thank you for your suggestion. We have made revisions to our manuscript. Please see the relevant part.

Reviewer 4 Report
It is an article on a topic of interest in the pharmaceutical field. There are only minimal details to consider.
1) It would be very convenient to include a scheme in which the routes of penetration of the drug through the skin are represented.
2) To delve further into the factors that affect the penetration of drugs through the skin.
3) It would be very convenient to include a general table that summarizes each of the generations of transdermal systems and examples.
In general, English is quite good.
Author Response
Reviewer #4:
It is an article on a topic of interest in the pharmaceutical field. It would be very convenient to include a scheme in which the routes of penetration of the drug through the skin are represented.
Response: Thank you for your positive comments.
1.To delve further into the factors that affect the penetration of drugs through the skin.
Response: Thank you for your suggestion. We have added relevant content to the manuscript. Please see Page 2 and Page 3.
2.It would be very convenient to include a general table that summarizes each of the generations of transdermal systems and examples.
Response: Thank you for your suggestion. We have added a general table that summarizes each of the generations of transdermal systems and examples.

Round 2
Reviewer 1 Report
no comment